# Towards Understanding Acceleration Tradeoff between Momentum and Asynchrony in Nonconvex Stochastic Optimization

**Tianyi Liu**
School of Industrial and System Engineering
Georgia Institute of Technology
Atlanta, GA 30332
tliu341@gatech.edu

**Shiyang Li**
Harbin Institue of Technology
lsydevin@gmail.com

**Jianping Shi**
Sensetime Group Limited
shijianping@sensetime.com

**Enlu Zhou**[*]
School of Industrial and System Engineering
Georgia Institute of Technology
Atlanta, GA 30332
enlu.zhou@isye.gatech.edu

**Tuo Zhao**[†]
School of Industrial and System Engineering
Georgia Institute of Technology
Atlanta, GA 30332
tuo.zhao@isye.gatech.edu

## Abstract

Asynchronous momentum stochastic gradient descent algorithms (Async-MSGD) have been widely used in distributed machine learning, e.g., training large collaborative filtering systems and deep neural networks. Due to current technical limit, however, establishing convergence properties of Async-MSGD for these highly complicated nonoconvex problems is generally infeasible. Therefore, we propose to analyze the algorithm through a simpler but nontrivial nonconvex problems — streaming PCA. This allows us to make progress toward understanding Aync-MSGD and gaining new insights for more general problems. Specifically, by exploiting the diffusion approximation of stochastic optimization, we establish the asymptotic rate of convergence of Async-MSGD for streaming PCA. Our results indicate a fundamental tradeoff between asynchrony and momentum: To ensure convergence and acceleration through asynchrony, we have to reduce the momentum (compared with Sync-MSGD). To the best of our knowledge, this is the first theoretical attempt on understanding Async-MSGD for distributed nonconvex stochastic optimization. Numerical experiments on both streaming PCA and training deep neural networks are provided to support our findings for Async-MSGD.

---
[*]Home Page: http://enluzhou.gatech.edu
[†]Home Page: https://www2.isye.gatech.edu/ tzhao80/

# 1  Introduction

Modern machine learning models trained on large data sets have revolutionized a wide variety of domains, from speech and image recognition (Hinton et al., 2012; Krizhevsky et al., 2012) to natural language processing (Rumelhart et al., 1986) to industry-focused applications such as recommendation systems (Salakhutdinov et al., 2007). Training these machine learning models requires solving large-scale nonconvex optimization. For example, to train a deep neural network given $n$ observations denoted by $\{(x_i, y_i)\}_{i=1}^n$, where $x_i$ is the $i$-th input feature and $y_i$ is the response, we need to solve the following empirical risk minimization problem,

$$\min_\theta \mathcal{F}(\theta) := \frac{1}{n} \sum_{i=1}^n \ell(y_i, f(x_i, \theta)), \tag{1}$$

where $\ell$ is a loss function, and $f$ is a neural network function/operator associated with parameter $\theta$.

Thanks to significant advances made in GPU hardware and training algorithms, we can easily train machine learning models on a GPU-equipped machine. For example, we can solve (1) using the popular momentum stochastic gradient descent (MSGD, Robbins and Monro (1951); Polyak (1964)) algorithm. Specifically, at the $t$-th iteration, we uniformly sample $i$ (or a mini-batch) from $(1, ..., n)$, and then take

$$\theta^{(k+1)} = \theta^{(k)} - \eta \nabla \ell(y_i, f(x_i, \theta^{(k)})) + \mu(\theta^{(k)} - \theta^{(k-1)}), \tag{2}$$

where $\eta$ is the step size parameter and $\mu \in [0, 1)$ is the parameter for controlling the momentum. Note that when $\mu = 0$, (2) is reduced to the vanilla stochastic gradient descent (VSGD) algorithm. Many recent empirical results have demonstrated the impressive computational performance of MSGD. For example, finishing a 180-epoch training with a moderate scale deep neural network (ResNet, 1.7 million parameters, He et al. (2016)) for CIFAR10 ($50,000$ training images in resolution $32 \times 32$) only takes hours with a NVIDIA Titan XP GPU.

For even larger models and datasets, however, solving (1) is much more computationally demanding and can take an impractically long time on a single machine. For example, finishing a 90-epoch ImageNet-1k (1 million training images in resolution $224 \times 224$) training with large scale ResNet (around 25.6 million parameters) on the same GPU takes over 10 days. Such high computational demand of training deep neural networks necessitates the training on distributed GPU cluster in order to keep the training time acceptable.

In this paper, we consider the "parameter server" approach (Li et al., 2014), which is one of the most popular distributed optimization frameworks. Specifically, it consists of two main ingredients: First, the model parameters are globally shared on multiple servers nodes. This set of servers are called the parameter servers. Second, there can be multiple workers processing data in parallel and communicating with the parameter servers. The whole framework can be implemented in either synchronous or asynchronous manner. The synchronous implementations are mainly criticized for the low parallel efficiency, since the servers always need to wait for the slowest worker to aggregate all updates within each iteration.

To circumvent this issue, practitioners have resorted to asynchronous implementations, which emphasize parallel efficiency by using potentially stale stochastic gradients for computation. Specifically, each worker in asynchronous implementations can process a mini-batch of data independently of the others, as follows: **(1)** The worker fetches from the parameter servers the most up-to-date parameters of the model needed to process the current mini-batch; **(2)** It then computes gradients of the loss with respect to these parameters; **(3)** Finally, these gradients are sent back to the parameter servers, which then updates the model accordingly. Since each worker communicates with the parameter servers independently of the others, this is called Asynchronous MSGD (Async-MSGD).

As can be seen, Async-MSGD is different from Sync-MSGD, since parameter updates may have occurred while a worker is computing its stochastic gradient; hence, the resulting stochastic gradients are typically computed with respect to outdated parameters. We refer to these as stale stochastic gradients, and its staleness as the number of updates that have occurred between its corresponding read and update operations. More precisely, at the $k$-th iteration, Async-MSGD takes

$$\theta^{(k+1)} = \theta^{(k)} - \eta \nabla \ell(y_i, f(x_i, \theta^{(k-\tau_k)})) + \mu(\theta^{(k)} - \theta^{(k-1)}), \tag{3}$$

where $\tau_k \in \mathbb{Z}_+$ denotes the delay in the system (usually proportional to the number of workers).

Understanding the theoretical impact of staleness is fundamental, but very difficult for distributed nonconvex stochastic optimization. Though there have been some recent papers on this topic, there are still significant gaps between theory and practice:

**(A)** They all focus on Async-VSGD (Lian et al., 2015; Zhang et al., 2015; Lian et al., 2016). Many machine learning models, however, are often trained using algorithms equipped with momentum such as Async-MSGD and Async-ADAM (Kingma and Ba, 2014). Moreover, there have been some results reporting that Async-MSGD sometimes leads to computational and generalization performance loss than Sync-MSGD. For example, Mitliagkas et al. (2016) observe that Async-MSGD leads to the generalization accuracy loss for training deep neural networks; Chen et al. (2016) observe similar results for Async-ADAM for training deep neural networks; Zhang and Mitliagkas (2018) suggest that the momentum for Async-MSGD needs to be adaptively tuned for better generalization performance.

**(B)** They all focus on analyzing convergence to a first order optimal solution (Lian et al., 2015; Zhang et al., 2015; Lian et al., 2016), which can be either a saddle point or local optimum. To better understand the algorithms for nonconvex optimization, machine learning researcher are becoming more and more interested in the second order optimality guarantee. The theory requires more refined characterization on how the delay affects escaping from saddle points and converging to local optima.

Unfortunately, closing these gaps of Async-MSGD for highly complicated nonconvex problems (e.g., training large recommendation systems and deep neural networks) is generally infeasible due to current technical limit. Therefore, we will study the algorithm through a simpler and yet nontrivial nonconvex problems — streaming PCA. This helps us to understand the algorithmic behavior of Async-MSGD better even in more general problems. Specifically, the stream PCA problem is formulated as

$$\max_v \ v^\top \mathbb{E}_{X \sim \mathcal{D}}[XX^\top]v \quad \text{subject to} \quad v^\top v = 1, \tag{4}$$

where $\mathcal{D}$ is an unknown zero-mean distribution, and the streaming data points $\{X_k\}_{k=1}^\infty$ are drawn independently from $\mathcal{D}$. This problem, though nonconvex, is well known as a strict saddle optimization problem over sphere (Ge et al., 2015), and its optimization landscape enjoys two geometric properties: (1) no spurious local optima and (2) negative curvatures around saddle points.

These nice geometric properties can also be found in several other popular nonconvex optimization problems, such as matrix regression/completion/sensing, independent component analysis, partial least square multiview learning, and phase retrieval (Ge et al., 2016; Li et al., 2016; Sun et al., 2016). However, little has been known for the optimization landscape of general nonconvex problems. Therefore, as suggested by many theoreticians, a strict saddle optimization problem such as streaming PCA could be a first and yet significant step towards understanding the algorithms. The insights we gain on such simpler problems shed light on more general nonconvex optimization problems. Illustrating through the example of streaming PCA, we intend to answer the fundamental question, which also arises in Mitliagkas et al. (2016):

*Does there exist a tradeoff between asynchrony and momentum*
*in distributed nonconvex stochastic optimization?*

The answer is "Yes". We need to reduce the momentum for allowing a larger delay. Roughly speaking, our analysis indicates that for streaming PCA, the delay $\tau_k$'s are allowed to asymptotically scale as

$$\tau_k \lesssim (1-\mu)^2/\sqrt{\eta}.$$

Moreover, our analysis also indicates that the asynchrony has very different behaviors from momentum. Specifically, as shown in Liu et al. (2018), the momentum accelerates optimization, when escaping from saddle points, or in nonstationary regions, but cannot improve the convergence to optima. The asynchrony, however, can always enjoy a linear speed up throughout all optimization stages. The linear speed-up can be understood as follows. We assume all the workers have similar performance, which is realistic when training Deep Neural Network where all GPUs are same. Async-MSGD works in a pipelining manner. Since we have more workers, Async-MSGD can complete $\tau$ updates in the one iteration time of MSGD. Thus, if we count $\tau$ updates of Async-MSGD as one iteration, the algorithm will enjoy a linear speed up (faster).

The main technical challenge for analyzing Async-MSGD comes from the complicated dependency caused by momentum and asynchrony. Our analysis adopts diffusion approximations of stochastic optimization, which is a powerful applied probability tool based on the weak convergence theory.

Existing literature has shown that it has considerable advantages when analyzing complicated stochastic processes (Kushner and Yin, 2003). Specifically, we prove that the solution trajectory of Async-MSGD for streaming PCA converges weakly to the solution of an appropriately constructed ODE/SDE. This solution can provide intuitive characterization of the algorithmic behavior, and establish the asymptotic rate of convergence of Async-MSGD. To the best of our knowledge, this is the first theoretical attempt of Async-MSGD for distributed nonconvex stochastic optimization.

**Notations**: For $1 \leq i \leq d$, let $e_i = (0, ..., 0, 1, 0, ..., 0)^\top$ (the $i$-th dimension equals to 1, others 0) be the standard basis in $\mathbb{R}^d$. Given a vector $v = (v^{(1)}, ..., v^{(d)})^\top \in \mathbb{R}^d$, we define the vector norm: $||v||^2 = \sum_j (v^{(j)})^2$. The notation $w.p.1$ is short for with probability one, $B_t$ is the standard Brownian Motion in $\mathbb{R}^d$, and $\mathbb{S}$ denotes the sphere of the unit ball in $\mathbb{R}^d$, i.e., $\mathbb{S} = \{v \in \mathbb{R}^d \,|\, ||v|| = 1\}$. $\dot{F}$ denotes the derivative of the function $F(t)$. $\asymp$ means asymptotically equal.

## 2 Async-MSGD and Optimization Landscape of Streaming PCA

Recall that we study Async-MSGD for the streaming PCA problem formulated as (4)

$$\max_v \ v^\top \mathbb{E}_{X \sim \mathcal{D}}[XX^\top]v \quad \text{subject to} \quad v^\top v = 1.$$

We apply the asynchronous stochastic generalized Hebbian Algorithm with Polyak's momentum (Sanger, 1989). Note that the serial/synchronous counterpart has been studied in Liu et al. (2018). Specifically, at the $k$-th iteration, given $X_k \in \mathbb{R}^d$ independently sampled from the underlying zero-mean distribution $\mathcal{D}$, Async-MSGD takes

$$v_{k+1} = v_k + \mu(v_k - v_{k-1}) + \eta(I - v_{k-\tau_k}v_{k-\tau_k}^\top)X_kX_k^\top v_{k-\tau_k}, \tag{5}$$

where $\mu \in [0, 1)$ is the momentum parameter, and $\tau_k$ is the delay. We remark that from the perspective of manifold optimization, (5) is essentially considered as the stochastic approximation of the manifold gradient with momentum in the asynchronous manner. Throughout the rest of this paper, if not clearly specified, we denote (5) as Async-MSGD for notational simplicity.

The optimization landscape of (4) has been well studied in existing literature. Specifically, we impose the following assumption on $\Sigma = \mathbb{E}[XX^\top]$.

**Assumption 1.** *The covariance matrix $\Sigma$ is positive definite with eigenvalues*

$$\lambda_1 > \lambda_2 \geq ... \geq \lambda_d > 0$$

*and associated normalized eigenvectors $v^1$, $v^2$, ..., $v^d$.*

Assumption 1 implies that the eigenvectors $\pm v^1$, $\pm v^2$, ..., $\pm v^d$ are all the stationary points for problem (4) on the unit sphere $\mathbb{S}$. Moreover, the eigen-gap ($\lambda_1 > \lambda_2$) guarantees that the global optimum $v^1$ is identifiable up to sign change, and moreover, $v^2$, ..., $v^{d-1}$ are $d-2$ strict saddle points, and $v^d$ is the global minimum (Chen et al., 2017).

## 3 Convergence Analysis

We analyze the convergence of the Async-MSGD by diffusion approximations. Our focus is to find the proper delay given the momentum parameter $\mu$ and the step size $\eta$. We first prove the global convergence of Async-MSGD using an ODE approximation. Then through more refined SDE analysis, we further establish the rate of convergence. Before we proceed, we impose the following mild assumption on the underlying data distribution:

**Assumption 2.** *The data points $\{X_k\}_{k=1}^\infty$ are drawn independently from some unknown distribution $\mathcal{D}$ over $R^d$ such that*

$$\mathbb{E}[X] = 0, \ \mathbb{E}[XX^\top] = \Sigma, \ ||X|| \leq C_d,$$

*where $C_d$ is a constant (possibly dependent on $d$).*

The boundedness assumption here can be further relaxed to a moment bound condition. The proof, however, requires much more involved truncation arguments, which is beyond the scope of this paper. Thus, we assume the uniform boundedness for convenience.

## 3.1 Global Convergence

We first show that the solution trajectory converges to the solution of an ODE. By studying the ODE, we establish the global convergence of Async-MSGD, and the rate of convergence will be established later. Specifically, we consider a continuous-time interpolation $V^{\eta,\tau}(t)$ of the solution trajectory of the algorithm: For $t \geq 0$, set $V^{\eta,\tau}(t) = v_k^{\eta,\tau}$ on the time interval $[k\eta, k\eta + \eta)$. Throughout our analysis, similar notations apply to other interpolations, e.g., $H^{\eta,\tau}(t), U^{\eta,\tau}(t)$.

To prove the weak convergence, we need to show the solution trajectory $\{V^{\eta,\tau}(t)\}$ must be tight in the Cadlag function space. In another word, $\{V^{\eta,\tau}(t)\}$ is uniformly bounded in $t$, and the maximum discontinuity (distance between two iterations) converges to 0, as shown in the following lemma:

**Lemma 1.** *Given $v_0 \in \mathbb{S}$, for any $k \leq O(1/\eta)$, we have $\|v_k\|^2 \leq 1 + O\left(\max_i \tau_i \eta / (1-\mu)^2\right)$. Specifically, given $\tau_k \lesssim (1-\mu)^2/\eta^{1-\gamma}$ for some $\gamma \in (0,1]$, we have*

$$\|v_k\|^2 \leq 1 + O(\eta^\gamma) \quad and \quad \|v_{k+1} - v_k\| \leq \frac{2C_d\eta}{1-\mu}.$$

The proof is provided in Appendix A.1. Roughly speaking, the delay is required to satisfy

$$\tau_k \lesssim (1-\mu)^2/\eta^{1-\gamma}, \ \forall k > 0,$$

for some $\gamma \in (0,1]$ such that the tightness of the trajectory sequence is kept. Then by Prokhorov's Theorem, this sequence $\{V^\eta(t)\}$ converges weakly to a continuous function. Please refer to Liu et al. (2018) for the prerequisite knowledge on weak convergence theory.

Then we derive the weak limit. Specifically, we rewrite Async-MSGD as follows:

$$v_{k+1} = v_k + \eta Z_k = v_k + \eta(m_{k+1} + \beta_k + \epsilon_k), \tag{6}$$

where

$$\epsilon_k = (\Sigma_k - \Sigma)v_{k-\tau_k} - v_{k-\tau_k}^\top(\Sigma_k - \Sigma)v_{k-\tau_k}v_{k-\tau_k},$$

$$m_{k+1} = \sum_{i=0}^{k} \mu^i [\Sigma v_{k-i-\tau_{k-i}} - v_{k-i-\tau_{k-i}}^\top \Sigma v_{k-i-\tau_{k-i}} v_{k-i-\tau_{k-i}}],$$

and $\beta_k = \sum_{i=0}^{k-1} \mu^{k-i} \left[ (\Sigma_i - \Sigma)v_{i-\tau_i} - v_i^\top (\Sigma_i - \Sigma)v_{i-\tau_i} v_{i-\tau_i} \right],$

As can bee seen in (6), the term $m_{k+1}$ dominates the update, and $\beta_k + \epsilon_k$ is the noise. Note that when we have momentum in the algorithm, $m_{k+1}$ is not a stochastic approximation of the gradient, which is different from VSGD. Actually, it is an approximation of $\widetilde{M}(v_k^\eta)$ and biased, where $\widetilde{M}(v) = \frac{1}{1-\mu}[\Sigma v - v^\top \Sigma vv]$. We have the following lemma to bound the approximation error.

**Lemma 2.** *For any $k > 0$, we have*

$$\|m_{k+1}^\eta - \widetilde{M}(v_k^\eta)\| \leq O\left(\eta \log(1/\eta)\right) + O\left(\frac{\tau_k \lambda_1 \eta}{(1-\mu)^2}\right), \quad w.p. \ 1.$$

Note that the first term in the above error bound comes from the momentum, while the second one is introduced by the delay. To ensure that this bound does not blow up as $\eta \to 0$, we have to impose a further requirement on the delay.

Given Lemmas 1 and 2, we only need to prove that the continuous interpolation of the noise term $\beta_k + \epsilon_k$ converges to 0, which leads to the main theorem.

**Theorem 3.** *Suppose for any $i > 0$, $v_{-i} = v_0 = v_1 \in \mathbb{S}$. When the delay in each step is chosen according to the following condition:*

$$\tau_k \lesssim (1-\mu)^2/(\lambda_1\eta^{1-\gamma}), \ \forall k > 0, \ for \ some \ \gamma \in (0,1],$$

*for each subsequence of $\{V^\eta(\cdot), \eta > 0\}$, there exists a further subsequence and a process $V(\cdot)$ such that $V^\eta(\cdot) \Rightarrow V(\cdot)$ in the weak sense as $\eta \to 0$ through the convergent subsequence, where $V(\cdot)$ satisfies the following ODE:*

$$\dot{V} = \frac{1}{1-\mu}[\Sigma V - V^\top \Sigma VV], \quad V(0) = v_0. \tag{7}$$

To solve ODE (7), we rotate the coordinate to decouple each dimension. Specifically, there exists an eigenvalue decomposition such that

$$\Sigma = Q\Lambda Q^\top, \quad \text{where} \ \Lambda = \text{diag}(\lambda_1, \lambda_2, ..., \lambda_d) \ \text{and} \ Q^\top Q = I.$$

Note that, after the rotation, $e_1$ is the optimum corresponding to $v_1$. Let $H^\eta(t) = Q^\top V^\eta(t)$, then we have as $\eta \to 0$, $\{H^\eta(\cdot), \eta > 0\}$ converges weakly to

$$H^{(i)}(t) = \Big( \sum_{i=1}^d \Big[ H^{(i)}(0) \exp\Big( \frac{\lambda_i t}{1-\mu} \Big) \Big]^2 \Big)^{-\frac{1}{2}} H^{(i)}(0) \exp\Big( \frac{\lambda_i t}{1-\mu} \Big), \ i = 1, ..., d.$$

Moreover, given $H^{(1)}(0) \neq 0$, $H(t)$ converges to $H^* = e_1$ as $t \to \infty$. This implies that the limiting solution trajectory of Async-MSGD converges to the global optima, given the delay $\tau_k \lesssim (1-\mu)^2/(\lambda_1 \eta^{1-\gamma})$ in each step.

Such an ODE approach neglects the noise and only considers the effect of the gradient. Thus, it is only a characterization of the mean behavior and is reliable only when the gradient dominates the variance throughout all iterations. In practice, however, we care about one realization of the algorithm, and the noise plays a very important role and cannot be neglected (especially near the saddle points and local optima, where the gradient has a relatively small magnitude). Moreover, since the ODE analysis does not explicitly characterize the order of the step size $\eta$, no rate of convergence can be established. In this respect, the ODE analysis is insufficient. Therefore, we resort to the SDE-based approach later for a more precise characterization.

### 3.2 Local Algorithmic Dynamics

The following SDE approach recovers the effect of the noise by rescaling and can provide a more precise characterization of the local behavior. The relationship between the SDE and ODE approaches is analogous to that between Central Limit Theorem and Law of Large Number.

• **Phase III: Around Global Optima**. We consider the normalized process
$$\{u_n^{\eta,\tau} = (h_n^{\eta,\tau} - e_1)/\sqrt{\eta}\}$$
around the optimal solution $e_1$, where $h_n^{\eta,\tau} = Q^\top v_n^{\eta,\tau}$. The intuition behind this rescaling is similar to "$\sqrt{N}$" in Central Limit Theorem.

We first analyze the error introduced by the delay after the above normalization. Let $D_n = H_{n+1} - H_n - \eta \sum_{i=0}^k \mu^{k-i}\{\Lambda_i H_i - H_i^\top \Lambda_i H_i H_i\}$ be the error . Then we have
$$u_{n+1} = u_n + \sqrt{\eta} \sum_{i=0}^k \mu^{k-i}\{\Lambda_i H_i - H_i^\top \Lambda_i H_i H_i\} + \frac{1}{\sqrt{\eta}} D_n.$$

Define the accumulative asynchronous error process as: $D(t) = \frac{1}{\sqrt{\eta}} \sum_{i=1}^{t/\eta} D_i$. To ensure the weak convergence, we prove that the continuous stochastic process $D(t)$ converges to zero as shown in the following lemma.

**Lemma 4.** *Given delay $\tau_k's$ satisfying*
$$\tau_k \asymp \frac{(1-\mu)^2}{(\lambda_1 + C_d)\eta^{\frac{1}{2}-\gamma}}, \ \forall k > 0,$$
*for some $\gamma \in (0, 0.5]$, we have for any $t$ fixed, $\lim_{\eta \to 0} D(t) \to 0$, a.s.*

Lemma 4 shows that after normalization, we have to use a delay smaller than that in Theorem 3 to control the noise. This justifies that the upper bound we derived from the ODE approximation is inaccurate for one single sample path.

We then have the following SDE approximation of the solution trajectory.

**Theorem 5.** *For every $k > 0$, the delay satisfies the following condition:*
$$\tau_k \asymp \frac{(1-\mu)^2}{(\lambda_1 + C_d)\eta^{\frac{1}{2}-\gamma}}, \ \forall k > 0, \text{ for some } \gamma \in (0, 0.5],$$
*as $\eta \to 0$, $\{U^{\eta,s,i}(\cdot)\}$ $(i \neq 1)$ converges weakly to a stationary solution of*
$$dU = \frac{(\lambda_i - \lambda_1)}{1-\mu} U dt + \frac{\alpha_{i,1}}{(1-\mu)} dB_t, \tag{8}$$
*where $\alpha_{i,j} = \sqrt{\mathbb{E}[(Y^{(i)})^2 (Y^{(j)})^2]}$ and $U^{\eta,s,i}(\cdot)$ is the $i$-th dimension of $U^{\eta,s}(\cdot)$.*

Theorem 8 implies that $\frac{(1-\mu)^2}{(\lambda_1 + C_d)\eta^{\frac{1}{2}-\gamma}}$ workers are allowed to work simultaneously. For notational simplicity, denote $\tau = \max_k \tau_k$ and $\phi = \sum_j \alpha_{1,j}^2$, which is bounded by the forth order moment of the data. Then the asymptotic rate of convergence is shown in the following proposition.

**Proposition 6.** *Given a sufficiently small $\epsilon > 0$ and*

$$\eta \asymp (1-\mu)\epsilon(\lambda_1 - \lambda_2)/\phi,$$

*there exists some constant $\delta \asymp \sqrt{\eta}$, such that after restarting the counter of time, if $\left(H^{\eta,1}(0)\right)^2 \geq 1 - \delta^2$, we allow $\tau$ workers to work simultaneously, where for some $\gamma \in (0, 0.5]$,*

$$\tau \asymp \frac{(1-\mu)^2}{(\lambda_1 + C_d)\eta^{\frac{1}{2}-\gamma}}, \quad \text{and we need } T_3 = \frac{(1-\mu)}{2(\lambda_1 - \lambda_2)} \log\left(\frac{(1-\mu)(\lambda_1 - \lambda_2)\delta^2}{(1-\mu)(\lambda_1 - \lambda_2)\epsilon - 2\eta\phi}\right)$$

*to ensure $\sum_{i=2}^d \left(H^{\eta,i}(T_3)\right)^2 \leq \epsilon$ with probability at least $3/4$.*

Proposition 6 implies that asymptotically, the effective iteration complexity of Async-MSGD enjoys a linear acceleration, i.e.,

$$N_3 \asymp \frac{T_3}{\tau\eta} \asymp \frac{(\lambda_1 + C_d)\phi^{\frac{1}{2}+\gamma}}{[(1-\mu)(\lambda_1 - \lambda_2)]^{\frac{3}{2}+\gamma}\epsilon^{\frac{1}{2}+\gamma}} \log\left(\frac{(1-\mu)(\lambda_1 - \lambda_2)\delta^2}{(1-\mu)(\lambda_1 - \lambda_2)\epsilon - 2\eta\phi}\right)$$

**Remark 7.** *Mitliagkas et al. (2016) conjecture that the delay in Async-SGD is equivalent to the momentum in MSGD. Our result, however, shows that this is not true in general. Specifically, when $\mu = 0$, Async-SGD yields an effective iterations of complexity:*

$$\widehat{N_3} \asymp \frac{(\lambda_1 + C_d)\phi^{\frac{1}{2}+\gamma}}{[(\lambda_1 - \lambda_2)]^{\frac{3}{2}+\gamma}\epsilon^{\frac{1}{2}+\gamma}} \log\left(\frac{(\lambda_1 - \lambda_2)\delta^2}{(\lambda_1 - \lambda_2)\epsilon - 2\eta\phi}\right),$$

*which is faster than that of MSGD (Liu et al. 2018):*

$$\widetilde{N_3} \asymp \frac{\phi}{\epsilon(\lambda_1 - \lambda_2)^2} \cdot \log\left(\frac{(\lambda_1 - \lambda_2)\delta^2}{(\lambda_1 - \lambda_2)\epsilon - 2\eta\phi}\right).$$

*Thus, there exists fundamental difference between these two algorithms.*

• **Phase II: Traverse between Stationary Points**. For Phase II, we study the algorithmic behavior once Async-MSGD has escaped from saddle points. During this period, since the noise is too small compared to the large magnitude of the gradient, the update is dominated by the gradient, and the influence of the noise is negligible. Thus, the ODE approximation is reliable before it enters the neighborhood of the optimum. The upper bound $\tau \lesssim (1-\mu)^2/\lambda_1\eta^{1-\gamma}$ we find in Section 3.1 works in this phase. Then we have the following proposition:

**Proposition 8.** *After restarting the counter of time, given $\eta \asymp \epsilon(\lambda_1 - \lambda_2)/\phi$, $\delta \asymp \sqrt{\eta}$, we can allow $\tau$ workers to work simultaneously, where for some $\gamma \in (0, 1]$,*

$$\tau \asymp \frac{(1-\mu)^2}{\lambda_1\eta^{1-\gamma}}, \quad \text{and we need } T_2 = \frac{(1-\mu)}{2(\lambda_1 - \lambda_2)} \log\left(\frac{1-\delta^2}{\delta^2}\right)$$

*such that $\left(H^{\eta,1}(T_2)\right)^2 \geq 1 - \delta^2$.*

Proposition 8 implies that asymptotically, the effective iteration complexity of Async-MSGD enjoys a linear acceleration by a factor $\tau$, i.e.,

$$N_2 \asymp \frac{T_2}{\tau\eta} \asymp \frac{\lambda_1\phi^\gamma}{2(1-\mu)(\lambda_1 - \lambda_2)^{1+\gamma}\epsilon^\gamma} \log\left(\frac{1-\delta^2}{\delta^2}\right).$$

• **Phase I: Escaping from Saddle Points**. At last, we study the algorithmic behavior around saddle points $e_j$, $j \neq 1$. Similarly to Phase I, the gradient has a relatively small magnitude, and noise is the key factor to help the algorithm escape from the saddles. Thus, an SDE approximation need to be derived. Define $\{u_n^{s,\eta} = (h_n^{s,\eta} - e_i)/\sqrt{\eta}\}$ for $i \neq 1$. By the same SDE approximation technique used in Section 3.2, we obtain the following theorem.

**Theorem 9.** *Condition on the event that $h_k^\eta - e_j \asymp \sqrt{\eta}$ for $k = 1, 2....$ Then for $i \neq j$, if for any $k$, the delay satisfies the following condition:*

$$\tau_k \asymp \frac{(1-\mu)^2}{(\lambda_1 + C_d)\eta^{\frac{1}{2}-\gamma}}, \quad \forall k > 0,$$

*for some $\gamma \in (0, 0.5]$, $\{U^{\eta,i}(\cdot)\}$ converges weakly to a solution of*

$$dU = \frac{(\lambda_i - \lambda_j)}{1-\mu} U dt + \frac{\alpha_{i,j}}{(1-\mu)} dB_t.$$

Here $h_k^\eta - e_j \asymp \sqrt{\eta}$ is only a technical assumption. When $(h_k^\eta - e_j)/\sqrt{\eta}$ is large, MSGD has escaped from the saddle point $e_j$, which is out of Phase I. In this respect, this assumption does not cause any issue.

We further have the following proposition:

**Proposition 10.** *Given a pre-specified $\nu \in (0,1)$, $\eta \asymp \epsilon(\lambda_1 - \lambda_2)/\phi$, and $\delta \asymp \sqrt{\eta}$, we allow $\tau$ workers to work simultaneously, where for some $\gamma \in (0, 0.5]$,*

$$\tau \asymp \frac{(1-\mu)^2}{(\lambda_1 + C_d)\eta^{\frac{1}{2}-\gamma}}, \quad \text{and we need } T_1 = \frac{1-\mu}{2(\lambda_1 - \lambda_2)} \log\left(2\frac{(1-\mu)\eta^{-1}\delta^2(\lambda_1 - \lambda_2)}{\Phi^{-1}\left(\frac{1+\nu}{2}\right)^2 \alpha_{12}^2} + 1\right)$$

*such that $(H^{\eta,2}(T_1))^2 \leq 1 - \delta^2$ with probability at least $1 - \nu$, where $\Phi(x)$ is the CDF of the standard normal distribution.*

Proposition 10 implies that asymptotically, the effective iteration complexity of Async-MSGD enjoys a linear acceleration, i.e.,

$$N_1 \asymp \frac{T_1}{\eta\tau} \asymp \frac{(\lambda_1 + C_d)\phi^{\frac{1}{2}+\gamma}}{2(1-\mu)(\lambda_1 - \lambda_2)^{\frac{3}{2}+\gamma}\epsilon^{\frac{1}{2}+\gamma}} \log\left(2\frac{(1-\mu)\eta^{-1}\delta^2(\lambda_1 - \lambda_2)}{\Phi^{-1}\left(\frac{1+\nu}{2}\right)^2 \alpha_{12}^2} + 1\right).$$

**Remark 11.** *We briefly summarize here: (1) There is a trade-off between the momentum and asynchrony. Specifically, to guarantee the convergence, delay must be chosen according to :*

$$\tau \asymp \frac{(1-\mu)^2}{(\lambda_1 + C_d)\eta^{\frac{1}{2}-\gamma}},$$

*for some $\gamma \in (0, 0.5]$. Then Async-MSGD asymptotically achieves a linear speed-up compared to MSGD. (2) Momentum and asynchrony have fundamental difference. With proper delays, Async-SGD achieves a linear speed-up in the third phase, while momentum cannot improve the convergence.*

## 4  Numerical Experiments

We present numerical experiments for both streaming PCA and training deep neural networks to demonstrate the tradeoff between the momentum and asynchrony. The experiment on streaming PCA verify our theory in Section 3, and the experiments on training deep neural networks verify that our theory, though trimmed for Streaming PCA, gains new insights for more general problems.

### 4.1  Streaming PCA

We first provide a numerical experiment to show the tradeoff between the momentum and asynchrony in streaming PCA. For simplicity, we choose $d = 4$ and the covariance matrix $\Sigma = \text{diag}\{4, 3, 2, 1\}$. The optimum is $(1, 0, 0, 0)$. We compare the performance of Async-MSGD with different delays and momentum parameters. Specifically, we start the algorithm at the saddle point $(0, 1, 0, 0)$ and set $\eta = 0.0005$. The algorithm is run for 100 times.

Figure 1 shows the average optimization error obtained by Async-MSGD with $\mu = 0.7, 0.8, 0.85, 0.9, 0.95$ and delays from 0 to 100. Here, the shade is the error bound. We see that for a fixed $\mu$, Async-MSGD can achieve similar optimization error to that of MSGD when the delay is below some threshold. We call it the optimal delay. As can be seen in Fig 1, the optimal delays for $\mu = 0.7, 0.8, 0.85, 0.9, 0.95$ are $120, 80, 60, 30, 10$ respectively. This indicates that there is a clear tradeoff between the asynchrony and momentum which is consistent with our theoretical analysis. We remark that the difference among Async-MSGD with different $\mu$ when $\tau = 0$ is due to the fact that the momentum hurts convergence, as shown in Liu et al. (2018).

### 4.2  Deep Neural Networks

We then provide numerical experiments for comparing different number workers and choices of momentum in training a 32-layer hyperspherical residual neural network (SphereResNet34) using the CIFAR-100 dataset for a 100-class image classification task. We use a computer workstation with 8 Titan XP GPUs. We choose a batch size of 128. 50k images are used for training, and the rest 10k are used for testing. We repeat each experiment for 10 times and report the average. We

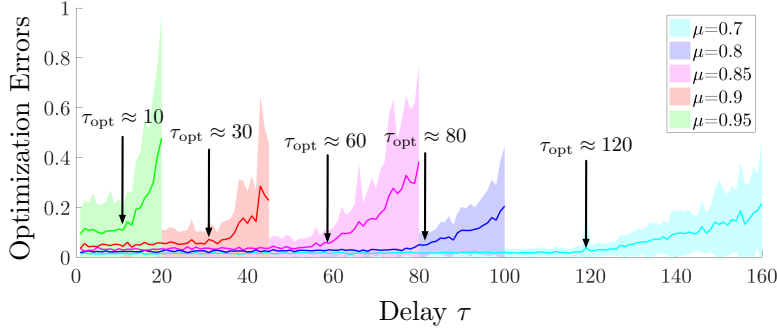

Figure 1: *Comparison of Async-MSGD with different momentum and delays. For $\mu = 0.7, 0.8, 0.85, 0.9, 0.95$, the optimal delay's are $\tau = 120, 80, 60, 30, 10$ respectively. This suggests a clear tradeoff between the asynchrony and momentum.*

choose the initial step size as $0.2$. We decrease the step size by a factor of $0.2$ after $60$, $120$, and $160$ epochs. The momentum parameter is tuned over $\{0.1, 0.3, 0.5, 0.7, 0.9\}$. More details on the network architecture and experimental settings can be found in He et al. (2016) and Liu et al. (2017). We repeat all experiments for 10 times, and report the averaged results.

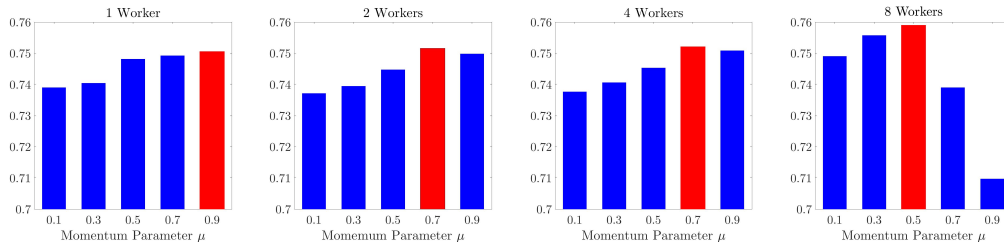

Figure 2: *The average validation accuracies of ResNet34 versus the momentum parameters with different numbers of workers. We can see that the optimal momentum decreases, as the number of workers increases.*

Figure 2 shows that the validation accuracies of ResNet34 under different settings. We can see that for one single worker $\tau = 1$, the optimal momentum parameter is $\mu = 0.9$; As the number of workers increases, the optimal momentum decreases; For 8 workers $\tau = 8$, the optimal momentum parameter is $\mu = 0.5$. We also see that $\mu = 0.9$ yields the worst performance for $\tau = 8$. This indicates a clear tradeoff between the delay and momentum, which is consistent with our theory.

## 5 Discussions

We remark that though our theory helps explain some phenomena in training DNNs, there still exist some gaps: (1) The optimization landscapes of DNNs are much more challenging than that of our studied streaming PCA problem. For example, there might exist many bad local optima and high order saddle points. How Async-MSGD behaves in these regions is still largely unknown; (2) Our analysis based on the diffusion approximations requires $\eta \to 0$. However, the experiments actually use relatively large step sizes at the early stage of training. Though we can expect large and small step sizes share some similar behaviors, they may lead to very different results; (3) Our analysis only explains how Async-MSGD minimizes the population objective. For DNNs, however, we are more interested in generalization accuracies. We will leave these open questions for future investigation.

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
