[Supplementary Material]

# A   Proof of the main results

## A.1   Proof of Lemma 1

*Proof.* First, if we assume $\{v_k\}$ is uniformly bounded by 2, we then have

$$v_{k+1} - v_k = \mu(v_k - v_{k-1}) + \eta\{\Sigma_k v_{k-\tau_k} - v_{k-\tau_k}^\top \Sigma_k v_{k-\tau_k} v_{k-\tau_k}\},$$

$$\Longrightarrow v_{k+1} - v_k = \sum_{i=0}^{k} \mu^{k-i}\eta\{\Sigma_i v_{i-\tau_i} - v_{i-\tau_i}^\top \Sigma_i v_{i-\tau_i} v_{i-\tau_i}\},$$

$$\Longrightarrow \|v_{k+1} - v_k\|_2 \le C_\delta \frac{\eta}{1-\mu},$$

where $C_\delta = \sup_{\|v\|\le 2, \|X\|\le C_d} \|XX^\top v - v^\top XX^\top vv\| \le 2C_d$. Thus, the jump $v_{k+1} - v_k$ is bounded. Next, we show the boundedness assumption on $v$ can be taken off. In fact, with an initialization on $\mathbb{S}$ (the sphere of the unit ball), the algorithm is bounded in a much smaller ball of radius $1 + O(\eta^\gamma)$.

Recall $\delta_{k+1} = v_{k+1} - v_k$. Let's consider the difference between the norm of two iterates,
$\Delta_k = \|v_{k+1}\|^2 - \|v_k\|^2 = \|\delta_{k+1}\|^2 + 2v_k^\top \delta_{k+1}$

$\Delta_{k+1} - \Delta_k = \|\delta_{k+2}\|^2 + 2v_{k+1}^\top \delta_{k+2} - \|\delta_{k+1}\|^2 - 2v_k^\top \delta_{k+1}$

$= \|\delta_{k+2}\|^2 - \|\delta_{k+1}\|^2 + 2\mu v_{k+1}^\top \delta_{k+1} + 2\eta v_{k+1}^\top [\Sigma_{k+1} v_{k+1-\tau_{k+1}} - v_{k+1-\tau_{k+1}}^\top \Sigma_{k+1} v_{k+1-\tau_{k+1}} v_{k+1-\tau_{k+1}}] - 2v_k^\top \delta_{k+1}$

$= \|\delta_{k+2}\|^2 - \|\delta_{k+1}\|^2 + 2\mu v_{k+1}^\top \delta_{k+1} + 2\eta v_{k+1-\tau_{k+1}}^\top [\Sigma_{k+1} v_{k+1-\tau_{k+1}} - v_{k+1-\tau_{k+1}}^\top \Sigma_{k+1} v_{k+1-\tau_{k+1}} v_{k+1-\tau_{k+1}}] +$

$\quad 2\eta[v_{k+1} - v_{k+1-\tau_{k+1}}]^\top [\Sigma_{k+1} v_{k+1-\tau_{k+1}} - v_{k+1-\tau_{k+1}}^\top \Sigma_{k+1} v_{k+1-\tau_{k+1}} v_{k+1-\tau_{k+1}}] - 2v_k^\top \delta_{k+1}$

$\le \|\delta_{k+2}\|^2 - \|\delta_{k+1}\|^2 + 2\mu v_k^\top \delta_{k+1} + 2\mu\|\delta_{k+1}\|^2 + 2\eta v_{k+1-\tau_{k+1}}^\top \Sigma_{k+1} v_{k+1-\tau_{k+1}}(1 - v_{k+1-\tau_{k+1}}^\top v_{k+1-\tau_{k+1}})$

$- 2v_k^\top \delta_{k+1} + \frac{C_\delta^2}{1-\mu}\tau_{k+1}\eta^2$

$= |\delta_{k+2}\|^2 + \mu\|\delta_{k+1}\|^2 - (1-\mu)(\|\delta_{k+1}\|^2 + 2v_k^\top \delta_{k+1}) + \frac{C_\delta^2}{1-\mu}\tau_{k+1}\eta^2$

$\quad + 2\eta v_{k+1-\tau_{k+1}}^\top \Sigma_{k+1} v_{k+1-\tau_{k+1}}(1 - v_{k+1-\tau_{k+1}}^\top v_{k+1-\tau_{k+1}})$

$= \|\delta_{k+2}\|^2 + \mu\|\delta_{k+1}\|^2 - (1-\mu)\Delta_k + 2\eta v_{k+1-\tau_{k+1}}^\top \Sigma_{k+1} v_{k+1-\tau_{k+1}}(1 - v_{k+1-\tau_{k+1}}^\top v_{k+1-\tau_{k+1}}) + \frac{C_\delta^2}{1-\mu}\tau_{k+1}\eta^2$

$\le \|\delta_{k+2}\|^2 + \mu\|\delta_{k+1}\|^2 - (1-\mu)\Delta_k + \frac{C_\delta^2}{1-\mu}\tau_{k+1}\eta^2, \qquad \text{when } 1 \le \|v_{k+1-\tau_{k+1}}\| \le 2.$

Let $\kappa = \inf\{i : \|v_{i+1}\| > 1\}$, then

$$\Delta_{\kappa+1} \le (1+\mu)(\frac{C_\delta}{1-\mu})^2\eta^2 + \mu\Delta_\kappa + \frac{C_\delta}{1-\mu}\tau_{\kappa+1}\eta^2.$$

Moreover, if $1 < \|v_{\kappa+i-\tau_{k+i}}\| \le 2$ holds for $i = 1, ..., n < \frac{t}{\eta}$, we have

$$\Delta_{\kappa+i} \le (1+\mu)(\frac{C_\delta}{1-\mu})^2\eta^2 + \mu\Delta_{\kappa+i-1}$$

$$\le \frac{1+\mu}{1-\mu}(\frac{C_\delta}{1-\mu})^2\eta^2 + \frac{C_\delta}{(1-\mu)^2}(\max_k \tau_k)\eta^2 + \mu^i\Delta_\kappa.$$

Thus,

$$\|v_{\kappa+n+1}\|^2 = \|v_\kappa\|^2 + \sum_{i=0}^{n} \Delta_{\kappa+i}$$

$$\le 1 + \frac{1}{1-\mu}\Delta_k + \frac{t}{\eta}\frac{1+\mu}{1-\mu}(\frac{C_\delta}{1-\mu})^2\eta^2 + \frac{t}{\eta}\frac{C_\delta}{(1-\mu)^2}(\max_k \tau_k)\eta^2$$

$$\le 1 + O(\frac{(\max_k \tau_k)\eta}{(1-\mu)^2}).$$

In other words, when $\eta$ is very small, and $\tau_k \asymp (1-\mu)^2/(\eta^{1-\gamma}))$, we cannot go far from $\mathbb{S}$ and the assumption that $\|v\| \le 2$ can be removed. $\qquad\square$

## A.2 Proof of Lemma 2

*Proof.* To prove the inequality, we decompose the error (left-hand) into two parts:
$$\|m_{k+1}^\eta - \widetilde{M}(v_k^\eta)\| \le \|m_{k+1}^\eta - \widetilde{M}(v_{k-\tau_k}^\eta)\| + \|\widetilde{M}(v_{k-\tau_k}^\eta) - \widetilde{M}(v_k^\eta)\|,$$
where the first term on the right is the error caused by the noise while the second term is that introduce by the asynchrony. We first bound the second term. In fact, it can be easily bounded by the Lipschitz continuity. Here the Lipschitz constant of $\widetilde{M}$ is $\lambda_1/(1-\mu)$, then we have:

$$\|\widetilde{M}(v_{k-\tau_k}^\eta) - \widetilde{M}(v_k^\eta)\| \le \frac{\lambda_1}{1-\mu}\|v_{k-\tau_k}^\eta - v_k^\eta\|$$

$$\le \frac{\lambda_1}{1-\mu}\tau_k C\eta/(1-\mu)$$

$$= O(\tau_k \lambda_1 \eta/(1-\mu)^2).$$

Next we are going to bound the first term. Since this can be now viewed as no-delay case, we can use the same method as in Appendix B.2 in Liu et al. (2018). Since $\frac{1}{1-\mu} = \sum_{i=0}^\infty \mu^i$, there exists $N(\eta) = \log_\mu(1-\mu)\eta$ such that $\sum_{i=N(\eta)}^\infty \mu^i < \eta$. When $k > N(\eta)$, write $m_k$ and $\widetilde{M}(v_k)$ into summations:

$$m_{k+1} = \sum_{i=0}^k \mu^i[\Sigma v_{k-i-\tau_{k-i}} - v_{k-i-\tau_{k-i}}^\top \Sigma v_{k-i-\tau_{k-i}} v_{k-i-\tau_{k-i}}]$$

$$= \sum_{i=0}^{N(\delta)} \mu^i[\Sigma v_{k-i-\tau_{k-i}} - v_{k-i-\tau_{k-i}}^\top \Sigma v_{k-i-\tau_{k-i}} v_{k-i-\tau_{k-i}}]$$

$$+ \sum_{i=N(\delta)+1}^k \mu^i[\Sigma v_{k-i-\tau_{k-i}} - v_{k-i-\tau_{k-i}}^\top \Sigma v_{k-i-\tau_{k-i}} v_{k-i-\tau_{k-i}}],$$

and
$$\widetilde{M}(v_k - \tau_k) = \frac{1}{1-\mu}[\Sigma v_{k-\tau_k} - v_{k-\tau_k}^\top \Sigma v_{k-\tau_k} v_{k-\tau_k}]$$

$$= \sum_{i=0}^{N(\delta)} \mu^i[\Sigma v_{k-\tau_k} - v_{k-\tau_k}^\top \Sigma v_{k-\tau_k} v_{k-\tau_k}] + \sum_{i=N(\delta)+1}^\infty \mu^i[\Sigma v_{k-\tau_k} - v_{k-\tau_k}^\top \Sigma v_{k-\tau_k} v_{k-\tau_k}].$$

Note that $\|v_{k+1} - v_k\| \le C\eta$, where $C = \frac{C_\delta}{1-\mu}$ is a constant. Then we have

$$\max_{i=0,1,...,N(\eta)}\|v_{k-i-\tau_{k-i}} - v_{k-\tau_k}\| \le \frac{C_\delta}{1-\mu}N(\eta)\eta + 2\frac{C_\delta}{1-\mu}\max_i \tau_i\eta.$$

They by Lipschitz continuity, for $i = 0, 1, ..., N(\delta)$, we have
$$\|\Sigma v_{k-\tau_k} - v_{k-\tau_k}^\top \Sigma v_{k-\tau_k} v_{k-\tau_k} - \Sigma v_{k-i-\tau_{k-i}} + v_{k-i-\tau_{k-i}}^\top \Sigma v_{k-i-\tau_{k-i}} v_{k-i-\tau_{k-i}}\|$$

$$\le \frac{\lambda_1 C_\delta}{1-\mu}N(\eta)\eta + 2\frac{\lambda_1 C_\delta}{1-\mu}\max_i \tau_i\eta.$$

Then
$$\left\|\sum_{i=0}^{N(\delta)} \mu^i\{[\Sigma v_{k-i} - v_{k-i}^\top \Sigma v_{k-i} v_{k-i}] - [\Sigma v_k - v_k^\top \Sigma v_k v_k]\}\right\| \le \frac{KCN(\eta)\eta}{1-\mu}$$

$$\le \frac{C_\delta}{(1-\mu)^2}N(\eta)\eta + 2\frac{C_\delta}{(1-\mu)^2}\max_i \tau_i\eta.$$

Since $\Sigma v_k - v_k^\top \Sigma v_k v_k$ is uniformly bounded by $C$ w.p.1, both $\sum_{i=N(\delta)+1}^k \mu^i[\Sigma v_{k-i-\tau_{k-i}} - v_{k-i-\tau_{k-i}}^\top \Sigma v_{k-i-\tau_{k-i}} v_{k-i-\tau_{k-i}}]$ and $\sum_{i=N(\delta)+1}^\infty \mu^i[\Sigma v_{k-tau_k} - v_{k-tau_k}^\top \Sigma v_{k-tau_k} v_{k-tau_k}]$ are bounded by $C\eta$. Thus,

$$\|m_{k+1} - \widetilde{M}(v_{k-tau_k})\| \le \frac{C_\delta}{(1-\mu)^2}N(\eta)\eta + 2\frac{C_\delta}{(1-\mu)^2}\max_i \tau_i\eta + 2C\eta$$

$$= O(\eta\log\frac{1}{\eta}) + O(\tau_k \lambda_1 \eta/(1-\mu)^2) \quad w.p.1.$$

For $k < N(\eta)$, following the same approach, we can bound $\|m_{k+1} - \widetilde{M}(v_k)\|$ by the same bound
. $\qquad \square$

### A.3 Proof of Theorem 3

*Proof.* The proof technique is the fixed-state-chain method introduced by Liu et al. (2018). Please refer to our Arxiv version[3] for more details. $\square$

### A.4 Proof of Lemma 4

*Proof.* Define $G_j(h) = \Lambda_j h - h^\top \Lambda_j hh = \Lambda h - h^\top \Lambda hh + X_j X_j^\top h - h^\top X_j X_j^\top hh$, which is smooth and bounded, thus Lipschitz. The Lipschitz constant is determined by $\Lambda$ and the data $X$. Since $X$ is bounded by Assumption 2, for any $j > 0$, we have
$$||G_j(h') - G_j(h'')|| \leq (C_d + \lambda_1)||h' - h''||.$$
Then we have:
$$||D_k|| = \eta || \sum_{j=0}^{k} \mu^{k-i}(G_j(H_j) - G_j(H_{j-s}))||$$
$$\leq \eta \sum_{j=0}^{k} \mu^{k-i} L_d ||H_j - H_{j-\tau_j}||$$
$$\leq \sum_{j=0}^{k} \mu^{k-i} L_d \tau_j C_\delta \frac{\eta^2}{1-\mu}$$
$$\leq C_\delta \frac{L_d \max_j \tau_j \eta^2}{(1-\mu)^2} = o(\eta^{3/2}).$$
Then from the definition of $D(t)$, we know $D(t) \to 0, a.s.$ $\square$

### A.5 Proof of Theorem 5

*Proof.* The proof method follows the proof of Theorem 4.1 in Liu et al. (2018). The detail proof is very involved and out of our major concern. Please refer to the Arxiv version for more details. $\square$

### A.6 Proof of Proposition 6

*Proof.* Since we restart our record time, we assume here the algorithm is initialized around the global optimum $e_1$. Thus, we have $\sum_{i=2}^{d}(U^{\eta,i}(0))^2 = \eta^{-1}\delta^2 < \infty$. Since $U^{\eta,i}(t)$ approximates to $U^{(i)}(t)$ in this neighborhood, and the second moment of $U^{(i)}(t)$ is: For $i \neq 1$,
$$\mathbb{E}\left(U^{(i)}(t)\right)^2 = \frac{\alpha_{i1}^2}{2(1-\mu)(\lambda_1 - \lambda_i)} + \left(\left(U^{(i)}(0)\right)^2 - \frac{\alpha_{i1}^2}{2(1-\mu)(\lambda_1 - \lambda_i)}\right) \exp\left[-2\frac{(\lambda_1 - \lambda_i)t}{1-\mu}\right],$$
by Markov inequality, we have:
$$\mathbb{P}\left(\sum_{i=2}^{d}\left(H_\eta^{(i)}(T_3)\right)^2 > \epsilon\right) \leq \frac{\mathbb{E}\left(\sum_{i=2}^{d}\left(H_\eta^{(i)}(T_3)\right)^2\right)}{\epsilon} = \frac{\mathbb{E}\left(\sum_{i=2}^{d}\left(U^{\eta,i}(T_3)\right)^2\right)}{\eta^{-1}\epsilon}$$
$$\approx \frac{1}{\eta^{-1}\epsilon}\sum_{i=2}^{d}\frac{\alpha_{i1}^2}{2(1-\mu)(\lambda_1 - \lambda_i)}\left(1 - \exp\left(-2\frac{(\lambda_1 - \lambda_i)T_3}{1-\mu}\right)\right)$$
$$+ \left(U^{\eta,i}(0)\right)^2 \exp\left[-2\frac{(\lambda_1 - \lambda_i)T_3}{1-\mu}\right]$$
$$\leq \frac{1}{\eta^{-1}\epsilon}\left(\frac{\phi}{2(1-\mu)(\lambda_1 - \lambda_2)}\left(1 - \exp\left(-2\frac{(\lambda_1 - \lambda_d)T_3}{1-\mu}\right)\right)\right.$$
$$\left. + \eta^{-1}\delta^2 \exp\left[-2\frac{(\lambda_1 - \lambda_2)T_3}{1-\mu}\right]\right)$$
$$\leq \frac{1}{\eta^{-1}\epsilon}\left(\frac{\phi}{2(1-\mu)(\lambda_1 - \lambda_2)} + \eta^{-1}\delta^2 \exp\left[-2\frac{(\lambda_1 - \lambda_2)T_3}{1-\mu}\right]\right).$$

To guarantee $\frac{1}{\eta^{-1}\epsilon}\left(\frac{\phi}{2(1-\mu)(\lambda_1-\lambda_2)}+\eta^{-1}\delta^2\exp\left[-2\frac{(\lambda_1-\lambda_2)T_3}{1-\mu}\right]\right)\leq\frac{1}{4}$, we have:

$$T_3 = \frac{1-\mu}{2(\lambda_1-\lambda_2)}\log\left(\frac{(1-\mu)(\lambda_1-\lambda_2)\delta^2}{(1-\mu)(\lambda_1-\lambda_2)\epsilon - 2\eta\phi}\right).$$

$\square$

### A.7 Proof of Proposition 10

*Proof.* Recall that Theorem 9 holds when $u_k^\eta = (h_k^\eta - e_2)/\sqrt{\eta}$ is bounded. Thus, if $(H_\eta^{(2)}(T_1))^2 \leq 1-\delta^2$ holds at some time $T_1$, the algorithm has successfully escaped the saddle point. We approximate $U^{\eta,1}(t)$ by the limiting process approximation, which is normal distributed at time $t$. As $\eta \to 0$, by simple manipulation, we have

$$\mathbb{P}\left((H^{\eta,2}(T_1))^2 \leq 1-\delta^2\right) = \mathbb{P}\left((U^{\eta,2}(T_1))^2 \leq \eta^{-1}(1-\delta^2)\right).$$

We then prove $P\left(\left|U^{\eta,1}(T_1)\right| \geq \eta^{-\frac{1}{2}}\delta\right) \geq 1-\nu$. At time t, $U^{\eta,1}(t)$ approximates to a normal distribution with mean 0 and variance $\frac{\alpha_{12}^2}{2(1-\mu)(\lambda_1-\lambda_2)}\left[\exp\left(2\frac{(\lambda_1-\lambda_2)T_1}{1-\mu}\right)-1\right]$. Therefore, let $\Phi(x)$ be the CDF of $N(0,1)$, we have

$$\mathbb{P}\left(\frac{\left|U^{\eta,1}(T_1)\right|}{\sqrt{\frac{\alpha_{12}^2}{2(1-\mu)(\lambda_1-\lambda_2)}\left[\exp\left(2\frac{(\lambda_1-\lambda_2)T_1}{1-\mu}\right)-1\right]}} \geq \Phi^{-1}\left(\frac{1+\nu}{2}\right)\right) \approx 1-\nu,$$

which requires

$$\eta^{-\frac{1}{2}}\delta \leq \Phi^{-1}\left(\frac{1+\nu}{2}\right)\cdot\sqrt{\frac{\alpha_{12}^2}{2(1-\mu)(\lambda_1-\lambda_2)}\left[\exp\left(2\frac{(\lambda_1-\lambda_2)T_1}{1-\mu}\right)-1\right]}.$$

Solving the above inequality, we get

$$T_1 = \frac{(1-\mu)}{2(\lambda_1-\lambda_2)}\log\left(\frac{2\eta^{-1}\delta^2(1-\mu)(\lambda_1-\lambda_2)}{\Phi^{-1}\left(\frac{1+\nu}{2}\right)^2\alpha_{12}^2}+1\right).$$

$\square$

### A.8 Proof of Proposition 8

*Proof.* After Phase I, we restart our record time, i.e., $H^{\eta,1}(0) = \delta$. Then we obtain

$$\left(H^{\eta,1}(T_2)\right)^2 \approx \left(H^{(1)}(T_2)\right)^2 = \left(\sum_{j=1}^{d}\left(\left(H^{(j)}(0)\right)^2\exp\left(2\frac{\lambda_j}{1-\mu}T_2\right)\right)\right)^{-1}\left(H^{(1)}(0)\right)^2\exp\left(2\frac{\lambda_1}{1-\mu}T_2\right)$$

$$\geq \left(\delta^2\exp(2\frac{\lambda_1}{1-\mu}T_2)+(1-\delta^2)\exp(2\frac{\lambda_2}{1-\mu}T_2)\right)^{-1}\delta^2\exp(2\frac{\lambda_2}{1-\mu}T_2),$$

which requires

$$\left(\delta^2\exp(2\frac{\lambda_1}{1-\mu}T_2)+(1-\delta^2)\exp(2\frac{\lambda_2}{1-\mu}T_2)\right)^{-1}\delta^2\exp(2\frac{\lambda_1}{1-\mu}T_2) \geq \eta^{-1}(1-\delta^2).$$

Solving the above inequality, we get

$$T_2 = \frac{1-\mu}{2(\lambda_1-\lambda_2)}\log\frac{1-\delta^2}{\delta^2}.$$

$\square$

## Footnotes

[3] https://arxiv.org/abs/1806.01660