[Reviews · NeurIPS 2018]

Reviewer 1



This manuscript analyzes the asymptotic convergence of asynchronous momentum SGD (AMSGD) for streaming PCA. The fundamental claim [line 101 & 239] is that asymptotically, for streaming PCA, the delay tau is allowed to scale as (1 - mu)^2 / sqrt(eta), where mu is the step size and mu the momentum parameter. Major Comments ============== Before we discuss the proof, I think the introduction is somewhat misleading. In line 76, the authors point out previous work all focus on analyzing convergence to a first order optimal solution. The readers can be confused that this paper improved the results of previous work. However, the problems studies in those paper and streaming PCA are different. As the authors also pointed out, streaming PCA is a very special case that all first order optimal solution is either global optimal or strict saddle point, but the previous work you mentioned studied more general or harder problems. I would suggest you clarity this part clearly. The proof has a few steps. In high level, the structure of the theoretical analysis should be improved. Quite a few places of the presentation have inconsistency or undefined symbols. I will mention them when discussing the proof as below. 1) Show the norm of iterates and the distance between any consecutive iterates are bounded. 2) Rewrite the ascent direction as a sum of three terms: one only involves the expectation of data matrix Sigma (m_k), and other two (beta_k and epsilon_k) contains the sample deviation (Simga_k - Simga). The notation Simga_k was not defined, though I can infer it's X_k X_k^T. Besides, a small mistake: in line 162, equation (6), eta is a factor for all three terms m, beta and epsilon, but the momentum parameter mu wasn't rescaled. One question I have is how to preserve the norm of v_k, as no projection step is used. Though we can see it's upper bounded, but it is not straightforward for me to see why it's also lower bounded. 3) Ignore beta and epsilon, proof that {v_k} converges to the principal eigenvector of Sigma. To show that, the authors exploits the rotation under eigenvalue decomposition and reparameterized V to H. 4) Consider the stochastic version of ODE. I'm lost in this part, as many notations are not defined. For example, in Thm 5, line 204, what is Y? In Proposition 6, what is phi? I don't have a good assessment for this part due to limited understanding. Overall, I think this paper is solid work, but the current version might be preliminary for a publishment. The theoretical part needs to be clarified to bring a clear and convincing analysis. Minor Comments =============== Line 24, 30, 63: This manuscript is notation heavy. I understand you want to first state general AMSGD and then focus on streaming PCA. But if you can replace theta by v in the introduction, it will save you one parameter. Line 32: equation (3) should be (2). (2) is MSGD and (3) is AMSGD Line 171: taht --> that Line 385 (Appendix) The hyperef to one section didn't compile.

Reviewer 2



This work provides novel analysis techniques to understand Asynchronous Momentum Stochastic Gradient Descent through the particular non-convex problem of streaming PCA. Their analysis technique is novel but the paper seems to be "nearly the same" as another submission (4980) to the degree that abstract, introduction, and several theorems are exactly same. 4980 solves the problem of Momentum Stochastic Gradient Descent while this paper adds asynchrony to it, but that only changes one step in the analysis and the rest is just the same. Given one of the two papers, the other one just follows with the same theorems and results. The asynchronous case might just have been a corollary to the paper 4980. The experiments of the two papers are different though. But the novelty of the paper is in theory and not in experiments, so I wouldn't place much weightage on them being different. As it is that the space of non-convex problems is vast, this work (both 4980 and 1861) provides good intuition on the role of momentum for this particular non-convex optimization. It shows how momentum helps escape saddle points but slows down convergence near the global minima. They use an ODE approach to show that if not initialized at saddle points or minima, the algorithm converges to the global minima. Next, they use an SDE based approach to characterize the local behavior around the global minima. This paper uses some new analysis techniques and gives valuable insight into non-convex optimization, at least for this particular problem. I would recommend, that if accepted, the authors merge the contributions of 4980 as a corollary. Also, there are several works that have pointed out the similarity of asynchronous SGD with momentum SGD (e.g. "Asynchrony begets momentum" Mitliagkas et al. 2016) and the subsequent citations on asynchronous SGD analysis. Authors are recommended to clarify the assumptions on asynchronous SGD analysis in relation to those works.

Reviewer 3



In this paper, asynchronous momentum stochastic gradient descent algorithms (Async-MSGD) is analyzed for streaming PCA, which is a relatively simple nonconvex optimization problem. The analytical tool is diffusion approximation following Liu et al. 2018. Particularly, the authors show the tradeoff between asynchrony and momentum: momentum has to be reduced when the delay from asynchrony is large. The tradeoff sounds reasonable and intuitive. I like the empirical results on neural networks, which is repeated several times to overcome randomness. I have some comments below. The main theory part (section 3) is a bit hard to follow. The current version reads like a stack of lemmas and theorems, which lacks motivations and connections between them. For example, line 171, why given lemmas 1 and 2, we only need ...’’? Lemma 1 is saying that iterates are on sphere bounded by stepsize. Since stepsize indicates how far the iterates move for each step, is this conclusion nontrivial? The analysis extensively used results from Liu et al. 2018, and also some texts. In fact, after bounding the delay \tau, the analysis is almost identical to Liu et al. 2018. I would like the authors to clarify the contribution, and elaborate the difference except in methodology, or emphasize the significance and challenge on bounding \tau. There are several claims of the paper that I am not fully convinced. Line 105, the asynchrony...can always enjoy a linear speed up...’’ Please explain linear speed up. A relative issue is that the maximum delay is assumed to be the number of workers, like in Line 205. And the number of workers is directly used to linearly speedup iterations in line 212-213. I am not sure if these are standard assumption for theoretical analysis. In practice, if one worker is really slow, will the delay be larger? Line 216, please explain why Async-SGD is faster’’. It is hard to tell for me from the equations. ========================================================== I update my score after read the rebuttal, and thanks the author for making the efforts. The updated score is based on merging contributions of 4980 into 1861. The theoretical contribution of 1861 alone is relatively incremental given 4980. I am also not satisfied with the authors' answer to speedup, the reference only show linear speedup for two experiments and does not make a strong claim as the authors.

Reviewer 4



This is a note from the area chair handling submissions 1861 and 4980. These two papers were flagged as a possible dual submission by our automatic checker. The reviewers who read them also noticed very high overlap between the submissions, including motivation, technical tools and results. Given this very high overlap, the authors should have considered submitting a single version of the work, and not two papers which apparently violate NIPS dual submission policy, and are an ethical violation (see https://nips.cc/Conferences/2018/PaperInformation/AuthorGuidelines). Note that the policy allows NIPS program chairs to reject all NIPS submissions by all authors of dual submissions, not just those deemed dual submissions. If the authors can make a very convincing case as to why these two submissions are sufficiently different, they should do so in the rebuttal (e.g., why is 4980 not a special case of 1861 with tau=0). Note that differences in empirical evaluation, or minor differences in presentation or emphasis, will not be accepted as an explanation. Note that the papers may of course be also rejected on grounds of their content, and you should address all other comments by the reviewers. *** After the author response. I still think these two submissions are ethically dubious, but willing to accept 1861 which is more general than the other one. The authors should merge the results into a single paper.